# Macroparasite Communities with Special Attention to Invasive Helminths in European Eels *Anguilla anguilla* from Freshwaters and Brackish Lagoons of a Mediterranean Island

Anaïs Esposito [1,*], Jean-José Filippi [2], Charlotte Gerbaud [1,2], Quentin Godeaux [1], Rémi Millot [2], Paul-Jean Agostini [3], Camille Albertini [4], Eric Durieux [1,2], Joséphine Foata [1] and Yann Quilichini [1,*]

1   UMR 6134 CNRS-Université de Corse Pascal Paoli Sciences Pour l'Environnement, 20250 Corte, France
2   UAR 3514 CNRS-Université de Corse Pascal Paoli Plateforme Marine Stella Mare, 20620 Biguglia, France
3   Fédération Départementale de Pêche de la Corse, 20090 Ajaccio, France
4   Office Français de la Biodiversité, Service Départemental de Haute-Corse, 20401 Bastia, France
*   Correspondence: esposito_a@univ-corse.fr (A.E.); quilichini_y@univ-corse.fr (Y.Q.)

**Abstract:** An extensive survey of macroparasites in 320 European eel *Anguilla anguilla* (Linnaeus, 1758) was conducted in two brackish lagoons and eleven freshwater localities in the Mediterranean island of Corsica (France) between spring 2021 and winter 2021–2022. It resulted in the identification of nineteen parasites: two Monogea, four Digenea, one Copepoda, four Acanthocephala, three Cestoda, and five Nematoda, including the first geographical records, as Corsican freshwater sites were studied for the first time. The silvering stage was determined, and the eels were aged through otolithometry to compare parasite communities. Classic parasitology indices, a multivariate analysis, and an analysis of indicator values (IndVal) showed clear preferences towards the host's habitat and salinity. Seasonal variations were shown for several parasites. A dataset from the same two coastal lagoons was used to study the changes in the parasite communities over the last decade, and this showed an increase in the prevalence and abundance of three invasive helminth species: the Monogenea *Pseudodactylogyrus bini* (Kikuchi, 1929), *Pseudodactylogyrus anguillae* (Yin and Sproston, 1948) Gusev, 1965 and the Nematoda *Anguillicola crassus* Kuwahara, Niimi, and Itagaki, 1974. These pathogenic parasites were found in all sampled localities, except for the two Monogenea in the polyhaline-to-euhaline Urbino lagoon. It is thus advised that future management measures take into account the environmental preferences of the most concerning parasites.

**Keywords:** parasite; *Anguilla anguilla*; invasive species; *Anguillicola crassus*; *Pseudodactylogyrus*; Corsica

**Key Contribution:** An extensive survey of the critically endangered and prized *Anguilla anguilla* revealed the presence of the pathogenic invasive parasites *Anguillicola crassus* and *Pseudodactylogyrus* spp. in the little-studied Mediterranean island of Corsica.

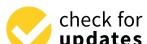



## 1. Introduction

The European eel *Anguilla anguilla* (Linnaeus, 1758) is a threatened catadromous teleost that migrates to the Sargasso Sea to reproduce. Leptocephalus larvae then migrate towards European and North African coasts, undergoing two metamorphoses: into glass eels upon reaching the continental shelf, and then into yellow eels after entering coastal, brackish, and freshwater habitats [1]. This commercial species has been declining for four decades, reaching 1–10% of its recruitment rate in the 1980s, and it was therefore assessed as critically endangered by the IUCN [2,3]. This decline has driven the European Commission to release a regulation [4] establishing a framework for the protection and sustainable use of the European eel stock. All European Union member states are called on to prepare an eel management plan per river basin aiming at the reduction of anthropogenic mortalities. *Anguilla anguilla* is vulnerable to several anthropogenic and natural threats: barriers to

migration, climate change and changes in oceanic currents, habitat loss and degradation, pollution, predation, legal and illegal exploitation and trade, non-parasitic invasive species, diseases, and parasites [3,5]. Parasites can be particularly linked to invasive species, as the latter can introduce and transmit their parasites to native species [6–8]. This issue is not to be overlooked in the context of management plans, as the occurrence of diseases and parasites influences the suitability of waterbodies for the restocking of juveniles and the growth of healthy spawners [9]. The parasite community of *A. anguilla* has drawn the attention of scientists throughout the host's distribution range, and more advanced and updated data are needed to better assess its evolution over time and amongst habitats. In this context, and on the basis of 148 research articles retrieved from Scopus and Web of Science concerning the parasitology of wild-caught eels since the year 2000, more than half of the studies concerning European eel parasites during the last two decades were conducted in Western Europe. Eels from France, Italy, the United Kingdom, Germany, and Spain received considerable attention. Mediterranean islands have been poorly studied, and only four articles could be retrieved during the period considered (in Corsica (France [10,11]) and Sardinia (Italy [12,13])). In Corsica, the topic was addressed in 2013 as part of the PhD thesis of Filippi [14], who provided a baseline for further studies on parasite communities of *A. anguilla* only in the Mediterranean coastal lagoons of the island. Studying Mediterranean islands, especially Corsica, is relevant to the knowledge of the European eel's parasite community, as the island is centrally located in this species' distribution range and encompasses a wide diversity of habitats. At the scale of the European eel distribution range, most studies were carried out in freshwater (e.g., [15–17]) and brackish (e.g., [18–20]) environments. Less attention has been paid to the marine environment (e.g., [9]). In Corsica, parasite communities of *A. anguilla* have only been studied in brackish coastal lagoons. Data for the freshwater environment are thus lacking despite the high density of rivers (3000 km of waterways for 1000 km of coastline) on the island.

The silvering stage is mostly disregarded and is unspecified in most studies. A few studies have focused on silver eels, yellow eels, and/or considered both (e.g., [9,11]). Elvers and glass eels are rarely studied. The lack of attention towards the silvering stage is unfortunate, as this process induces changes in the metabolism and behavior of the eel and has been shown to influence parasite communities. In eels, the growth rate has been shown to vary according to the habitat and sex, with faster growth in saline and brackish habitats than in freshwater habitats [21–23] and for females [24]. For this reason, the silvering stage and age are not always correlated, and individual age should also be considered when studying parasite communities, as well as individual length.

The swim bladder nematode *Anguillicola crassus* Kuwahara, Niimi, and Itagaki, 1974 has drawn a lot of attention from researchers, as have the branchial monogeneans *Pseudodactylogyrus bini* (Kikuchi, 1929) and *Pseudodactylogyrus anguillae* (Yin and Sproston, 1948) Gusev, 1965 to a lesser extent. Out of the 100 publications focused on one or two parasite species, 73 focused on *A. crassus*, 6 on *P. bini*, and 5 on *P. anguillae*. The introduction of these parasites to Western Europe resulted from the intercontinental transfer of live Japanese eels *Anguilla japonica* Temminck and Schlegel, 1846 [25–27]. These invasive parasites are considered serious pathogens, as *A. crassus* is known to impair the functioning of the swim bladder and thus threatens the success of transatlantic breeding migration [28–30]; furthermore, *P. bini* and *P. anguillae* are known to cause mortality (up to 90% if left untreated) and economic losses in aquaculture [29,31]. A few other parasites have attracted attention: Acantocephala *Acanthocephalus rhinensis* Amin, Thielen, Münderl, Taraschewskim, and Sures, 2008, *Paratenuisentis ambiguus* (Van Cleave, 1921) Bullock & Samuel, 1975 and *Pomphorhynchus laevis* (Zoega in Müller, 1776) Porta, 1908, the Digenea *Bucephalus anguillae* Spakulova, Macko, Berrilli, and Dezfuli, 2002, Nematoda *Contracaecum rudolphii* Hartwich, 1964, and *Paraquimperia tenerrima* (von Linstow, 1878) Baylis, 1934.

The aims of the present investigation are: (1) to fill in the gaps concerning knowledge of the *A. anguilla* parasite community in a Mediterranean island by examining, for the first time, freshwater European eels in Corsica; (2) to compare parasite communities between

freshwater and brackish habitats in Corsica through the use of a variety of statistical tools, and to compare them to continental regions and other islands throughout the host's distribution range; (3) to assess the distribution of fish parasites according to both biotic (silvering stage and age) and abiotic (habitat and season) factors; (4) to study the long-term variations in the parasite community on the basis of comparisons with lagoon community data from a decade ago [11]; and (5) to focus on species considered invasive in the European eel's distribution range and discuss this issue with regard to the management of this threatened resource.

## 2. Material and Methods

### 2.1. Sample Collection and Study Site

A total of 320 *A. anguilla* were taken from 13 different sampling sites across the island of Corsica (French Mediterranean, Figure 1, Table 1). Two brackish lagoons were sampled seasonally from spring 2021 to winter 2021–2022 by professional fishermen using fyke nets. Biguglia lagoon is the largest Corsican lagoon (14.5 km$^2$), and Urbino lagoon is the second largest (7.6 km$^2$). Both lagoons were selected for this study as representative of the eastern coast lagoons of Corsica and taking into account the availability of eel samples from fishermen. The lagoons differ with regard to their salinity range, Biguglia being oligohaline to polyhaline (7.3 PSU to 27.2 PSU) and Urbino being polyhaline to euhaline (35 PSU to 39 PSU) [32–34]. They also differ in terms of temperature (which is higher in Biguglia because of a maximum depth of 2 m, while the maximum depth in Urbino is 12 m) and nutrient concentration (Biguglia is eutrophic while Urbino is oligotrophic) [32,33,35]. Eleven freshwater sites were sampled from spring to fall 2021 via electrofishing in compliance with French legislation with the help of the Office Français de la Biodiversité (OFB) and of the Fédération de la Corse pour la Pêche et la Protection du Milieu Aquatique. Sampled watercourses were selected for their sufficient eel abundance on the basis of data obtained through the European Union Water Framework Directive fish monitoring conducted by the OFB. Samples were taken randomly among fishermen's catches and electrofishing catches, with a total length in the range 18–77 cm.

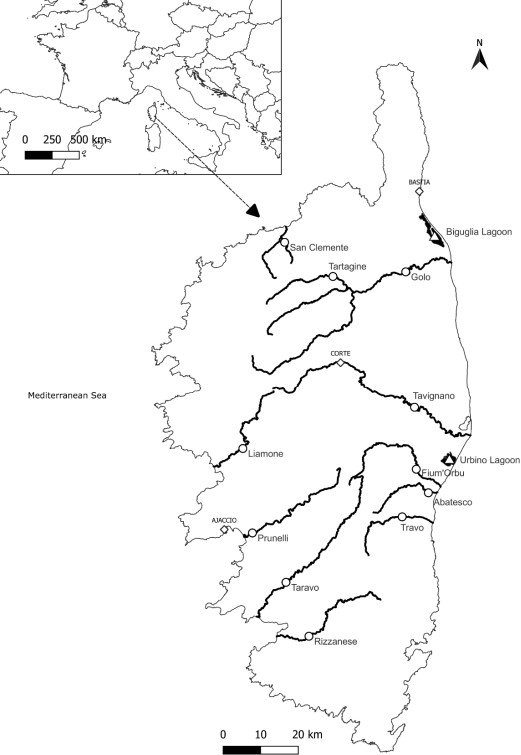

**Figure 1.** Sampling sites for eels in Corsica from the watercourses (circles) and both brackish lagoons, Biguglia and Urbino (triangles). Diamonds are the major cities in Corsica.

**Table 1.** Total length, weight, age, Fulton condition factor (mean ± standard deviation (minimum–maximum)), and silvering stage (determined following Durif, Guibert, and Elie 2009) for eels sampled in two brackish lagoons and eleven rivers in Corsica. I, undifferentiated resident; FII, female resident; FIII, pre-migrant female; FIV and FV, migrant females; MII, migrant male; N, number of eels.

| | | N | Total Length (mm) | Total Weight (g) | Age (a) | Fulton Condition Index | N I | N FII | N FIII | N FV | N MII |
|---|---|---|---|---|---|---|---|---|---|---|---|
| **Lagoon** | **Season** | | | | | | | | | | |
| Biguglia | Spring 2021 | 20 | 386 ± 54 (313–525) | 117 ± 63 (54–315) | 6 ± 1 (4–7) | 0.19 ± 0.02 (0.15–0.24) | 18 | 2 | 0 | 0 | 0 |
| | Summer 2021 | 20 | 409 ± 44 (355–517) | 126 ± 49 (80–287) | 6 ± 1 (5–7) | 0.18 ± 0.03 (0.09–0.21) | 14 | 3 | 0 | 0 | 3 |
| | Fall 2021 | 20 | 382 ± 29 (305–425) | 98 ± 26 (35–150) | 6 ± 1 (4–7) | 0.17 ± 0.02 (0.12–0.21) | 4 | 0 | 0 | 0 | 16 |
| | Winter 2021–2022 | 20 | 380 ± 23 (334–449) | 92 ± 18 (61–149) | 6 ± 1 (5–8) | 0.17 ± 0.01 (0.15–0.18) | 3 | 1 | 0 | 0 | 16 |
| Urbino | Spring 2021 | 20 | 421 ± 41 (341–510) | 105 ± 29 (58–184) | 8 ± 2 (6–13) | 0.14 ± 0.05 (0.09–0.34) | 8 | 9 | 0 | 1 | 2 |
| | Summer 2021 | 20 | 466 ± 121 (327–771) | 178 ± 159 (47–661) | 9 ± 3 (6–17) | 0.15 ± 0.02 (0.12–0.21) | 11 | 5 | 4 | 0 | 0 |
| | Fall 2021 | 20 | 432 ± 64 (344–597) | 149 ± 75 (59–379) | 8 ± 2 (6–14) | 0.17 ± 0.02 (0.14–0.22) | 10 | 7 | 0 | 0 | 3 |
| | Winter 2021–2022 | 20 | 379 ± 61 (307–580) | 94 ± 60 (40–319) | 8 ± 2 (5–14) | 0.16 ± 0.01 (0.13–0.19) | 15 | 1 | 0 | 1 | 3 |
| **River** | | | | | | | | | | | |
| Abatesco | | 15 | 368 ± 83 (278–541) | 123 ± 86 (41–295) | 11.1 ± 2.8 (7–16) | 0.21 ± 0.03 (0.18–0.27) | 12 | 3 | 0 | 0 | 0 |
| Fium'Orbo | | 15 | 320 ± 77 (256–582) | 79 ± 86 (25–377) | 8.3 ± 2.7 (5–17) | 0.18 ± 0.03 (0.15–0.26) | 12 | 0 | 1 | 0 | 2 |
| Golo | | 11 | 3717 ± 87 (234–504) | 126 ± 81 (48–280) | 9.3 ± 3.5 (5–15) | 0.24 ± 0.12 (0.17–0.58) | 7 | 2 | 1 | 0 | 1 |
| Liamone | | 15 | 332 ± 47 (282–479) | 89 ± 54 (45–273) | 8.3 ± 1.6 (6–13) | 0.22 ± 0.03 (0.19–0.26) | 14 | 0 | 0 | 0 | 1 |
| Prunelli | | 15 | 370 ± 62 (304–493) | 90 ± 47 (46–194) | 8.5 ± 1.8 (6–12) | 0.17 ± 0.02 (0.13–0.2) | 11 | 2 | 1 | 0 | 1 |

**Table 1.** *Cont.*

| | N | Total Length (mm) | Total Weight (g) | Age (a) | Fulton Condition Index | N I | N FII | N FIII | N FV | N MII |
|---|---|---|---|---|---|---|---|---|---|---|
| Rizzanese | 15 | 309 ± 31 (266–363) | 53 ± 19 (31–88) | 6.5 ± 1.1 (5–9) | 0.17 ± 0.02 (0.14–0.23) | 15 | 0 | 0 | 0 | 0 |
| San Clemente | 15 | 305 ± 17 (280–340) | 60 ± 13 (41–80) | 6.9 ± 1.0 (5–8) | 0.21 ± 0.04 (0.16–0.29) | 13 | 0 | 0 | 0 | 2 |
| Taravo | 15 | 295 ± 78 (181–406) | 60 ± 46 (8–154) | 7.2 ± 2.5 (3–12) | 0.18 ± 0.03 (0.13–0.23) | 15 | 0 | 0 | 0 | 0 |
| Tartagine | 14 | 386 ± 71 (266–512) | 117 ± 86 (29–321) | 7.4 ± 2.2 (3–11) | 0.18 ± 0.03 (0.15–0.24) | 11 | 1 | 1 | 0 | 1 |
| Tavignano | 15 | 335 ± 59 (259–502) | 74 ± 47 (26–217) | 7.5 ± 1.6 (5–11) | 0.18 ± 0.02 (0.15–0.21) | 13 | 1 | 0 | 0 | 1 |
| Travo | 15 | 309 ± 28 (259–367) | 59 ± 17 (28–93) | 9.1 ± 1.4 (7–12) | 0.20 ± 0.03 (0.16–0.24) | 11 | 0 | 0 | 0 | 4 |

### 2.2. Morphometry and Parasitological Examination

Eels were euthanized in compliance with French legislation, transported on ice in individual bags to the laboratory, and kept on ice until examination. Each eel was weighed to the nearest gram ($T_W$, in g), and the total length ($T_L$, in mm), the vertical and horizontal eye diameters, and the length of the pectoral fin were measured (in mm). Those external measurements were used to calculate a silvering index following the method described in Durif et al. [36] in order to classify the eels into six maturation stages. These stages corresponded to growth phases: undifferentiated resident (I), female resident (FII), pre-migrant female (FIII), migrant females (FIV and FV), and migrant male (MII). The body condition was measured through Fulton's condition factor ($K = 100 \cdot T_W \cdot T_L^{-3}$) (Table 1).

The gills, fins, stomach, intestine, gall bladder, swim bladder, spleen, and liver were placed in separate Petri dishes with physiological saline and examined under a stereomicroscope for parasites. The skin, mouth, and abdominal cavity were also checked for parasites. All parasites collected were preserved in 70% ethanol.

Additionally, data on parasite communities of eels from Biguglia and Urbino lagoons sampled from 2009 to 2012 were obtained from Filippi et al. (2013) [11] (Supplementary Table S1). Parasitological examination followed the same procedure as described earlier.

### 2.3. Otolithometry

For each European eel, both sagittal otoliths were removed and prepared following the International Council for the Exploration on the Sea (ICES) guidelines. The right sagitta was used for age estimation except if damaged during either extraction from the fish or the sectioning process or unreadable; in these cases, the left sagitta was used. The otolith was initially cleaned in 96% ethanol in a small vial up to 48 h (time depending on sagitta size). Otoliths were mounted in concave face-down position on microscope slides with thermoplastic glue (Crystalbond$^{TM}$509; Aremco$^{TM}$ products, New York, NY, USA) and ground or polished using an abrasive sheet (400, 800, and 1200 grit Wetordry$^{TM}$, 3M$^{TM}$) until the nucleus was reached. Otoliths were colored following current protocols [37] in order to improve otolith reading by increasing the contrast between the translucent and opaque growth bands. Thus, sections were etched with EDTA 5% for 5 min and subsequently stained with Toluidine Blue 5% for 2 min. These sections were examined and captured under a stereomicroscope (Zeiss Discovery V20, Zeiss, Oberkochen, Germany) connected to a camera (Sony XCD-U100CR, Sony, Tokyo, Japan). The age estimation was based on counting of the annuli, as described in ICES (2009) [38], by three independent readers.

### 2.4. Statistical Analysis

Parasite indices were calculated following the terminology of Bush et al. [39]: prevalence is the number of hosts infected with at least one individual of a particular parasite species divided by the number of hosts examined and expressed as a percentage; mean abundance is the total number of individuals of a given parasite species in a sample divided by the total number of hosts in that sample; and mean intensity is the total number of individuals of a given parasite species in a sample divided by the number of infected hosts in that sample.

The Shannon diversity index and the Berger–Parker dominance index were calculated at the individual level. These indices were used to compare parasite diversity and dominance between habitats through pairwise Wilcoxon tests. Pairwise Wilcoxon tests were also used to compare *Pseudodactylogyrus* spp. and *A. crassus* abundances between 2009–2012 and 2021–2022 for each season.

A Factor Analysis of Mixed Data (FAMD) was performed using the Factoshiny R package [40] by considering the most significant parasite species, total length, age, and Fulton index as quantitative variables and the habitat and season as qualitative variables to allow a better visualization of parasite communities.

Species are considered bio-indicators when their abundance and/or occurrence in a particular habitat are high [41]. An analysis of indicator values (IndVal) [42] was used

to combine the parasite species' relative abundance (specificity) and relative occurrence (fidelity) for a given variable. Three habitats were considered (both lagoons and freshwater) for this analysis, and then the seasons and the eels' silvering stage and age were studied for each habitat. A test for correlation using Kendall's tau showed that age and total length were significantly associated; thus, only age was selected for indicator value analysis. Specificity is the mean abundance of a parasite species in a given group of European eels divided by the same parasite abundance infecting all European eels. Fidelity is defined as the percentage of European eels in a given group infected by a given parasite species. The indicator values analysis's capacity to include both specificity and fidelity in the same index constitutes an advantage over classical statistical tests (e.g., ANOVA) when looking for indicator species in highly variable communities, such as parasites [41]. Calculations of IndVal and associated *p*-values (10,000 permutations) were conducted using the labdsv R package [43].

## 3. Results

### 3.1. Parasite Diversity in European Eels in Corsica

Nineteen macroparasites were identified in Corsican eels in 2021–2022 (Tables 2 and 3), including two Monogea, four Digenea, one Copepoda, four Acanthocephala, three Cestoda, and five Nematoda.

The most prevalent parasites in the oligohaline-to-polyhaline waters of Biguglia lagoon were the Monogenea *P. anguillae* and *P. bini*, the Copepoda *Ergasilus gibbus* von Nordmann, 1832 (both branchial parasites), and the Nematoda *A. crassus*, while eels from the polyhaline-to-haline Urbino lagoon were mainly infested with endoparasites: the Digenea *B. anguillae*, *Deropristis inflata* (Molin, 1859) Odhner, 1902, *Lecithochirium musculus* (Looss, 1907) Nasir and Diaz, 1971, the Acanthocephala *Southwellina hispida* (Van Cleave, 1925) Witenberg, 1932, and the Nematoda *Contracaecum* sp. Railliet and Henry, 1912. Concerning the rivers, eels were mainly infected with *P. anguillae*, *P. bini*, and *A. crassus* (both present in all freshwater sites) and the Nematoda *P. tenerrima* (present in half of the freshwater sites). *Contracaecum* sp. was found in four freshwater sites. *Bothriocephalus claviceps* (Goeze, 1782) Rudolphi, 1810 was found in two freshwater sites (Liamone and Taravo), with low abundance and prevalence. Parasite diversity, as assessed through the Shannon index, was the highest (*p*-value < 0.05) in Biguglia (0.56 ± 0.27), then in Urbino (0.47 ± 0.38), while freshwater parasite diversity (0.28 ± 0.34) was the lowest. Dominance followed the opposite trend, as it was significantly higher (*p*-value < 0.05) in freshwater sites (0.84 ± 0.18) than in the two lagoons (Biguglia, 0.76 ± 0.15; Urbino 0.79 ± 0.19). Concerning freshwater sites, parasite species' richness ranged from two species in the Tartagine to six species in the Taravo, in which the eels harbored an unidentified Digenea recovered exclusively from this site.

It should be emphasized that *P. anguillae* and *P. bini* were found in all the investigated sites except for Urbino lagoon, and *A. crassus* was found in all the investigated sites (Tables 2 and 3). Some individuals were found to harbor a strong load of these parasites: up to more than 1200 *Pseudodactylogyrus* (Biguglia lagoon, fall) and 41 *A. crassus* (Biguglia lagoon, spring).

**Table 2.** Composition of parasite communities of *Anguilla anguilla* in Corsican lagoons in 2021–2022, with information on prevalence, mean abundance (±standard deviation), mean intensity (±standard deviation), and diversity indices: the Shannon diversity index and Berger–Parker's dominance index; N, number of eels.

| | **Biguglia** | | | | | | | | | | | |
| --- | --- | --- | --- | --- | --- | --- | --- | --- | --- | --- | --- | --- |
| | **Spring 2021** | | | **Summer 2021** | | | **Fall 2021** | | | **Winter 2021–2022** | | |
| N | 20 | | | 20 | | | 20 | | | 20 | | |
| Shannon index | 0.71 ± 0.27 (0.32–1.47) | | | 0.6 ± 0.23 (0.06–1.17) | | | 0.47 ± 0.27 (0.05–1.17) | | | 0.47 ± 0.27 (0.03–0.93) | | |
| Berger–Parker dominance index | 0.7 ± 0.15 (0.41–0.92) | | | 0.72 ± 0.14 (0.43–0.99) | | | 0.83 ± 0.13 (0.49–0.99) | | | 0.81 ± 0.15 (0.53–1) | | |
| | Preva-lence | Mean abundance | Mean intensity | Preva-lence | Mean abundance | Mean intensity | Preva-lence | Mean abundance | Mean intensity | Preva-lence | Mean abundance | Mean intensity |
| Copepoda *Ergasilus gibbus* | 100 | 11.15 ± 6.75 | 11.15 ± 6.75 | 100 | 38.5 ± 32.13 | 38.5 ± 32.13 | 75 | 12.7 ± 15.2 | 16.93 ± 15.39 | 85 | 38.7 ± 49.86 | 33.76 ± 52.63 |
| Cestoda *Bothriocephalus claviceps* | 0 | - | - | 0 | - | - | 0 | - | - | 5 | 0.1 ± 0.45 | 2 ± - |
| *Proteocephalus macrocephalus* | 10 | 0.7 ± 2.7 | 7 ± 7.07 | 0 | - | - | 0 | - | - | 0 | - | - |
| Larvae indet. | 0 | - | - | 0 | - | - | 0 | - | - | 0 | - | - |
| Digenea *Bucephalus anguillae* | 0 | - | - | 0 | - | - | 0 | - | - | 0 | - | - |
| *Deropristis inflata* | 0 | - | - | 0 | - | - | 5 | 0.3 ± 1.34 | 6 ± - | 15 | 0.15 ± 0.37 | 1 ± 0 |
| *Lecithochirium musculus* | 0 | - | - | 0 | - | - | 0 | - | - | 0 | - | - |
| Monogenea *Pseudodactylogyrus anguillae* + *P. bini* | 100 | 35.85 ± 24.36 | 35.85 ± 24.36 | 100 | 126.65 ± 133.75 | 126.65 ± 133.75 | 100 | 161.5 ± 273.96 | 161.5 ± 273.96 | 100 | 131.25 ± 141.62 | 131.25 ± 141.62 |
| Acanthocephala *Solearhynchus rhytidotes* | 0 | - | - | 0 | - | - | 0 | - | - | 0 | - | - |
| *Southwellina hispida* (cystacanth) | 5 | 0.05 ± 0.22 | 1 ± - | 0 | - | - | 5 | 0.05 ± 0.22 | 1 ± - | 0 | - | - |
| *Telosentis exiguus* | 0 | - | - | 0 | - | - | 0 | - | - | 0 | - | - |
| Acanthocephala indet. | 0 | - | - | 0 | - | - | 0 | - | - | 0 | - | - |
| Nematoda *Anguillicola crassus* | 75 | 5.35 ± 9.52 | 7.13 ± 10.46 | 50 | 1.25 ± 1.94 | 2.5 ± 2.12 | 95 | 3.6 ± 2.72 | 3.79 ± 2.66 | 60 | 2.55 ± 2.12 | 4.25 ± 2.99 |
| *Contracaecum* sp. (encysted larvae) | 5 | 1.3 ± 5.81 | 26 ± - | 15 | 4.8 ± 14.63 | 32 ± 26.96 | 35 | 3.74 ± 10.83 | 11.83 ± 17.53 | 25 | 0.74 ± 1.79 | 3.5 ± 2.2 |

**Table 2.** *Cont.*

| | Urbino | | | | | | | | | | | |
|---|---|---|---|---|---|---|---|---|---|---|---|---|
| | Spring 2021 | | | Summer 2021 | | | Fall 2021 | | | Winter 2021–2022 | | |
| N | 20 | | | 20 | | | 20 | | | 20 | | |
| Shannon index | 0.26 ± 0.26 (0–0.69) | | | 0.43 ± 0.3 (0–0.86) | | | 0.62 ± 0.43 (0–1.78) | | | 0.55 ± 0.42 (0–1.18) | | |
| Berger–Parker dominance index | 0.88 ± 0.15 (0.5–1) | | | 0.8 ± 0.16 (0.52–1) | | | 0.72 ± 0.21 (0.29–1) | | | 0.75 ± 0.2 (0.47–1) | | |
| | Preva-lence | Mean abundance | Mean intensity | Preva-lence | Mean abundance | Mean intensity | Preva-lence | Mean abundance | Mean intensity | Preva-lence | Mean abundance | Mean intensity |
| **Copepoda** | | | | | | | | | | | | |
| *Ergasilus gibbus* | 0 | - | - | 0 | - | - | 0 | - | - | 0 | - | - |
| **Cestoda** | | | | | | | | | | | | |
| *Bothriocephalus claviceps* | 0 | - | - | 0 | - | - | 0 | - | - | 0 | - | - |
| *Proteocephalus macrocephalus* | 0 | - | - | 0 | - | - | 0 | - | - | 0 | - | - |
| Larvae indet. | 0 | - | - | 0 | - | - | 10 | 0.1 ± 0.31 | 1 ± 0 | 5 | 0.05 ± 0.22 | 1 ± - |
| **Digenea** | | | | | | | | | | | | |
| *Bucephalus anguillae* | 55 | 14.25 ± 44.4 | 25.91 ± 58.43 | 50 | 23.05 ± 48.82 | 46.1 ± 62.06 | 45 | 6.25 ± 19.47 | 13.89 ± 27.95 | 44 | 4.19 ± 7.03 | 9.57 ± 7.96 |
| *Deropristis inflata* | 25 | 1.35 ± 4.25 | 5.4 ± 7.64 | 0 | - | - | 55 | 6.65 ± 17.85 | 12.09 ± 23.09 | 78 | 34.83 ± 88.41 | 44.79 ± 98.7 |
| *Lecithochirium musculus* | 15 | 0.15 ± 0.37 | 1 ± 0 | 35 | 4.05 ± 9.71 | 11.57 ± 14.05 | 50 | 10.5 ± 15.26 | 21 ± 15.71 | 68 | 3.94 ± 6.68 | 5.77 ± 7.45 |
| **Monogenea** | | | | | | | | | | | | |
| *Pseudodactylogyrus anguillae + P. bini* | 0 | - | - | 0 | - | - | 0 | - | - | 0 | - | - |
| **Acanthocephala** | | | | | | | | | | | | |
| *Solearhynchus rhytidotes* | 0 | - | - | 0 | - | - | 10 | 0.1 ± 0.31 | 1 ± 0 | 0 | - | - |
| *Southwellina hispida* (cystacanth) | 11 | 0.32 ± 1.16 | 3 ± 2.83 | 55 | 2.4 ± 4.51 | 4.36 ± 5.41 | 55 | 3.15 ± 5.21 | 5.73 ± 5.95 | 0 | - | - |
| *Telosentis exiguus* | 0 | - | - | 0 | - | - | 0 | - | - | 5 | 0.1 ± 0.45 | 2 ± - |
| Acanthocephala indet. | 0 | - | - | 5 | 0.05 ± 0.22 | 1 ± - | 15 | 0.35 ± 1.14 | 2.33 ± 2.31 | 0 | - | - |
| **Nematoda** | | | | | | | | | | | | |
| *Anguillicola crassus* | 10 | 0.2 ± 0.7 | 2 ± 1.41 | 15 | 0.3 ± 0.8 | 2 ± 1 | 0 | - | - | 15 | 0.15 ± 0.37 | 1 ± 0 |
| *Contracaecum* sp. (encysted larvae) | 68 | 33.11 ± 52.44 | 48.38 ± 57.65 | 70 | 53.15 ± 137.43 | 75.93 ± 160.44 | 70 | 36.85 ± 68.12 | 52.64 ± 76.72 | 55 | 6.68 ± 12.15 | 12.7 ± 14.5 |

**Table 3.** Composition of parasite communities of *Anguilla anguilla* in Corsican rivers in 2021–2022, with information on prevalence, mean abundance (±standard deviation), mean intensity (±standard deviation), and diversity indices: the Shannon diversity index and Berger–Parker's dominance index; N, number of eels.

| | Abatesco | | | Fium'Orbo | | | Golo | | | Liamone | | |
|---|---|---|---|---|---|---|---|---|---|---|---|---|
| **N** | 15 | | | 15 | | | 11 | | | 15 | | |
| Shannon index | 0.53 ± 0.39 (0–1.33) | | | 0.16 ± 0.25 (0–0.69) | | | 0.14 ± 0.22 (0–0.64) | | | 0.55 ± 0.38 (0–1.01) | | |
| Berger–Parker dominance index | 0.7 ± 0.19 (0.33–1) | | | 0.91 ± 0.17 (0.5–1) | | | 0.94 ± 0.11 (0.67–1) | | | 0.76 ± 0.18 (0.5–1) | | |
| | Preva-lence | Mean abundance | Mean intensity | Preva-lence | Mean abundance | Mean intensity | Preva-lence | Mean abundance | Mean intensity | Preva-lence | Mean abundance | Mean intensity |
| **Copepoda** *Ergasilus gibbus* | 6.67 | 0.27 ± 1.03 | 4 ± - | 0 | - | - | 0 | - | - | 0 | - | - |
| **Cestoda** *Bothriocephalus claviceps* | 0 | - | - | 0 | - | - | 0 | - | - | 13.33 | 0.13 ± 0.35 | 1 ± 0 |
| **Digenea** Unidentified Digenea | 0 | - | - | 0 | - | - | 0 | - | - | 0 | - | - |
| **Monogenea** *Pseudodactylogyrus anguillae* + *P. bini* | 66.67 | 4.13 ± 4.41 | 6.2 ± 5.59 | 60 | 7.93 ± 17.58 | 13.22 ± 21.5 | 81.81 | 10 ± 13.51 | 12.22 ± 14.05 | 86.67 | 6.8 ± 10.51 | 7.85 ± 10.95 |
| **Nematoda** *Anguillicola crassus* | 53.33 | 2.67 ± 3.96 | 5 ± 4.24 | 53.33 | 1.33 ± 1.76 | 2.5 ± 1.69 | 36.36 | 1.18 ± 2.44 | 3.25 ± 3.3 | 53.33 | 1 ± 1.56 | 1.88 ± 1.73 |
| *Contracaecum* sp. (encysted larvae) | 0 | - | - | 6.67 | 0.47 ± 1.81 | 7 ± - | 7 | - | - | 6.67 | 0.33 ± 1.29 | 5 ± - |
| *Paraquimperia tenerrima* | 60 | 3.27 ± 4.03 | 5.44 ± 3.88 | 0 | - | - | 0 | - | - | 66.67 | 1.33 ± 1.8 | 2 ± 1.88 |
| *Spinitectus inermis* | 0 | - | - | 0 | - | - | 9.09 | 0.09 ± 0.3 | 1 ± - | 0 | - | - |
| *Pseudocapillaria tomentosa* | 0 | - | - | 0 | - | - | 0 | - | - | 0 | - | - |

| | Prunelli | | | Rizzanese | | | San Clemente | | | Taravo | | |
|---|---|---|---|---|---|---|---|---|---|---|---|---|
| **N** | 15 | | | 15 | | | 15 | | | 15 | | |
| Shannon index | 0.19 ± 0.28 (0–0.69) | | | 0.37 ± 0.26 (0–0.69) | | | 0.34 ± 0.33 (0–0.74) | | | 0.48 ± 0.42 (0–1.1) | | |
| Berger–Parker dominance index | 0.89 ± 0.17 (0.5–1) | | | 0.82 ± 0.18 (0.5–1) | | | 0.84 ± 0.17 (0.5–1) | | | 0.74 ± 0.24 (0.33–1) | | |
| | Preva-lence | Mean abundance | Mean intensity | Preva-lence | Mean abundance | Mean intensity | Preva-lence | Mean abundance | Mean intensity | Preva-lence | Mean abundance | Mean intensity |
| **Copepoda** *Ergasilus gibbus* | 0 | - | - | 0 | - | - | 0 | - | - | 0 | - | - |

**Table 3.** *Cont.*

| | Prevalence | Mean abundance | Mean intensity | Prevalence | Mean abundance | Mean intensity | Prevalence | Mean abundance | Mean intensity | Prevalence | Mean abundance | Mean intensity |
|---|---|---|---|---|---|---|---|---|---|---|---|---|
| **Cestoda** | | | | | | | | | | | | |
| *Bothriocephalus claviceps* | 0 | - | - | 0 | - | - | 0 | - | - | 6.67 | 0.07 ± 0.26 | 1 ± - |
| **Digenea** | | | | | | | | | | | | |
| Unidentified Digenea | 0 | - | - | 0 | - | - | 0 | - | - | 33.33 | 2 ± 6.39 | 6 ± 10.63 |
| **Monogenea** | | | | | | | | | | | | |
| *Pseudodactylogyrus anguillae* + *P. bini* | 80 | 14.47 ± 36.87 | 18.08 ± 40.73 | 93.33 | 9.73 ± 10.54 | 10.43 ± 10.57 | 80 | 4.47 ± 4.55 | 5.58 ± 4.42 | 73.33 | 12.87 ± 19.26 | 17.55 ± 20.72 |
| **Nematoda** | | | | | | | | | | | | |
| *Anguillicola crassus* | 40 | 0.93 ± 1.62 | 2.33 ± 1.86 | 66.67 | 1.73 ± 2.09 | 2.6 ± 2.07 | 53.33 | 1.8 ± 2.31 | 3.36 ± 2.13 | 26.67 | 0.6 ± 1.24 | 2.25 ± 1.5 |
| *Contracaecum* sp. (encysted larvae) | 0 | - | - | 0 | - | - | 0 | - | - | 6.67 | 0.07 ± 0.26 | 1 ± - |
| *Paraquimperia tenerrima* | 0 | - | - | 26.67 | 0.4 ± 0.74 | 1.5 ± 0.58 | 33.33 | 1 ± 1.92 | 3 ± 2.35 | 53.33 | 0.87 ± 1.19 | 1.63 ± 1.19 |
| *Spinitectus inermis* | 0 | - | - | 0 | - | - | 0 | - | - | 0 | - | - |
| *Pseudocapillaria tomentosa* | 6.67 | 0.27 ± 1.03 | 4 ± - | 0 | - | - | 0 | - | - | 0 | - | - |

| | **Tartagine** | | | **Tavignano** | | | **Travo** | | |
|---|---|---|---|---|---|---|---|---|---|
| **N** | 14 | | | 15 | | | 15 | | |
| Shannon index | 0 ± 0 (0–0) | | | 0.13 ± 0.24 (0–0.68) | | | 0.18 ± 0.32 (0–0.9) | | |
| Berger–Parker dominance index | 1 ± 0 (1–1) | | | 0.94 ± 0.1 (0.76–1) | | | 0.87 ± 0.16 (0.62–1) | | |
| | Prevalence | Mean abundance | Mean intensity | Prevalence | Mean abundance | Mean intensity | Prevalence | Mean abundance | Mean intensity |
| **Copepoda** | | | | | | | | | |
| *Ergasilus gibbus* | 0 | - | - | 0 | - | - | 0 | - | - |
| **Cestoda** | | | | | | | | | |
| *Bothriocephalus claviceps* | 0 | - | - | 0 | - | - | 0 | - | - |
| **Digenea** | | | | | | | | | |
| Unidentified Digenea | 0 | - | - | 0 | - | - | 0 | - | - |
| **Monogenea** | | | | | | | | | |
| *Pseudodactylogyrus anguillae* + *P. bini* | 35.71 | 29.57 ± 83.78 | 82.8 ± 131.53 | 80 | 11 ± 17.04 | 13.75 ± 18.12 | 40 | 2.6 ± 5.83 | 6.5 ± 8.04 |
| **Nematoda** | | | | | | | | | |
| *Anguillicola crassus* | 7.14 | 0.07 ± 0.27 | 1 ± - | 26.67 | 1.2 ± 3.34 | 4.5 ± 5.69 | 33.33 | 1.13 ± 1.77 | 3.4 ± 1.14 |
| *Contracaecum* sp. (encysted larvae) | 0 | - | - | 6.67 | 0.07 ± 0.26 | 1 ± - | 0 | - | - |
| *Paraquimperia tenerrima* | 0 | - | - | 0 | - | - | 20 | 0.93 ± 2.34 | 4.67 ± 3.51 |
| *Spinitectus inermis* | 0 | - | - | 0 | - | - | 0 | - | - |
| *Pseudocapillaria tomentosa* | 0 | - | - | 0 | - | - | 0 | - | - |

### 3.2. Analysis of Parasite Community of European Eels in Corsica

The FAMD showed a high variability of the *A. anguilla* parasite community according to the habitat (Figure 2, Biguglia lagoon or Urbino lagoon or rivers), as this variable was significantly well-represented in both dimensions considered. Besides the habitat, the variables contributing the most to the new axis were the total length, as eels from Urbino lagoon tended to be longer, the abundance of *S. hispida*, *L. musculus*, *B. anguillae*, and *Contracaecum* sp., which seem characteristic of Urbino lagoon, the abundance of *Pseudodactylogyrus* spp. and *E. gibbus*, which were characteristic of Biguglia lagoon, and the abundance of *P. tenerrima*, which seemed characteristic of freshwater sites. Age and Fulton's condition factor were also well-projected and showed that eels from Biguglia tended to be the youngest and eels from Urbino tended to have the lowest condition factor (Figure 2, Table 1). A correlation analysis showed no negative relationship between infection with *Pseudodactylogyrus* spp. or *A. crassus* and body condition (Fulton's condition index). However, a significant relationship (*p*-value < 0.05) was observed between *Pseudodactylogyrus* load and eel age, but only in Biguglia lagoon. Parasite communities in all three habitats are thus distinct despite the common presence of invasive parasites. Among 19 parasites, 10 were identified as having a significant indicator value (*p*-value < 0.05, Table 4) through randomization. Three species were identified as indicators of Biguglia lagoon: *Pseudodactylogyrus* spp., *E. gibbus*, and *A. crassus*. The specificity of *Pseudodactylogyrus* spp. was high, but it was not at the maximum, as these parasites were also recovered from the totality of the freshwater sites. The fidelity for this parasite was at the maximum, as all examined eels were found to be infected. The specificity for *E. gibbus* was also very high, but it was not at the maximum, as this copepod was found once in the Abatesco, and its fidelity was quite high. *Anguillicola crassus* was recovered from both lagoons and all freshwater sites, but both its specificity and fidelity were markedly higher for Biguglia. Using an analysis of indicator values, six species were found to be indicators of Urbino lagoon: *B. anguillae*, *D. inflata*, *L. musculus*, *S. hispida* (cystacanth), unidentified Acanthocephala, and *Contracaecum* sp. (encysted larvae). The specificities of *B. anguillae*, *L. musculus*, and unidentified Acanthocephala were at the maximum in Urbino, as they were not recovered from the other sites.

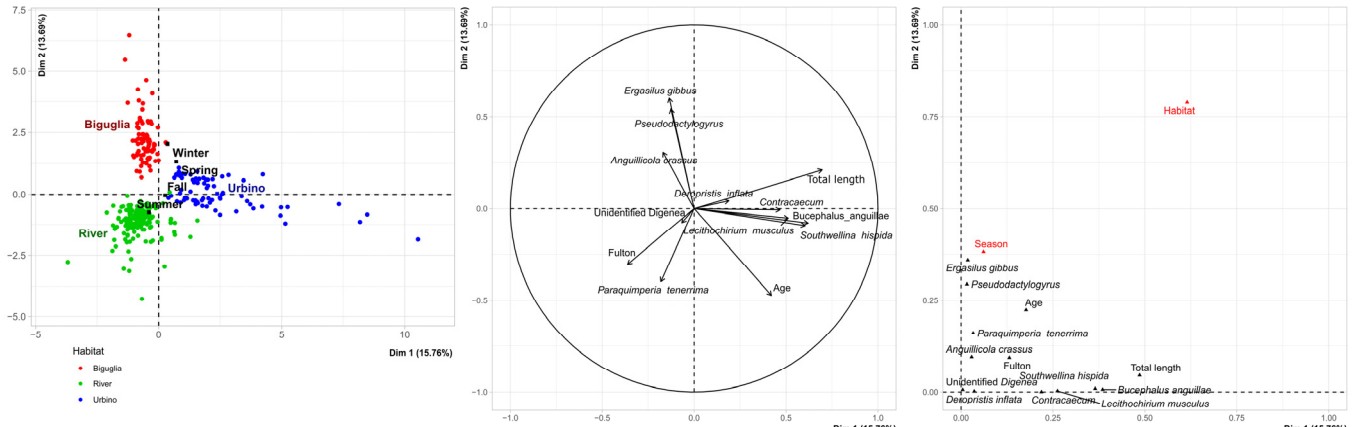

**Figure 2.** FAMD of *Anguilla anguilla* parasite communities from two brackish lagoons and eleven rivers in Corsica.

Only one parasite was identified as an indicator of rivers: the Nematoda *P. tenerrima*. The specificity for this species in Corsican freshwater sites was at the maximum as it was not recovered from either of the studied lagoons, and fidelity was quite low.

**Table 4.** Specificity (%), fidelity (%), and indicator value of ten parasite species relative to their habitats. ** *p*-value < 0.01; **** *p*-value ≤ 0.0001.

| Habitat | Biguglia Lagoon | Urbino Lagoon | Rivers |
|---|---|---|---|
| Number of eels | 78 | 72 | 159 |
| Monogenea *Pseudodactylogyrus anguillae + P. bini* | 91.9–100.0.0 (91.9) **** | 0.0–0.0 (0.0) | 8.1–71.1 (5.8) |
| Digenea *Bucephalus anguillae* | 0.0–0.0 (0.0) | 100.0.0–47.2 (47.2) **** | 0.0–0.0 (0.0) |
| *Deropristis inflata* | 1.1–5.1 (0.1) | 98.9–36.1 (35.7) **** | 0.0–0.0 (0.0) |
| *Lecithochirium musculus* | 0.0–0.0 (0.0) | 100.0.0–38.9 (38.9) **** | 0.0–0.0 (0.0) |
| Copepoda *Ergasilus gibbus* | 99.9–89.7 (89.6) **** | 0.0–0.0 (0.0) | 0.1–0.6 (0.0) |
| Acanthocephala *Southwellina hispida* | 1.6–2.5 (0.0) | 98.4–31.3 (30.7) **** | 0.0–0.0 (0.0) |
| Acanthocephala indet. | 0.0–0.0 (0.0) | 100.0.0–5.5 (5.5) ** | 0.0–0.0 (0.0) |
| Nematoda *Anguillicola crassus* | 68.7–69.2 (47.6) **** | 3.7–9.7 (0.4) | 27.6–41.5 (11.5) |
| *Contracaecum* sp. | 7.0–17.9 (1.3) | 92.7–66.7 (61.8) **** | 0.2–2.5 (0.0) |
| *Paraquimperia tenerrima* | 0.0–0.0 (0.0) | 0.0—0.0 (0.0) | 100.0.0–24.5 (24.5) **** |

*3.3. Seasonal Variations of Parasite Communities in Coastal Lagoons*

Seasonal variations were observed in brackish coastal lagoons. Parasite species' richness in brackish lagoons ranged from four species in Biguglia during summer 2021 to eight species in Urbino during fall 2021. In Biguglia, *Pseudodactylogyrus* species seemed to reach a low in spring, with abundances markedly lower compared to other seasons. *Ergasilus gibbus* infection oscillated, with summer and winter abundances notably higher than spring and fall abundances. The randomization test identified this Copepoda as an indicator (*p*-value < 0.05, Table 5) of summer. The specificity for this species was higher in summer than any other seasons, and fidelity was at the maximum, as all eels were found to be infected.

**Table 5.** Specificity (%), fidelity (%), and indicator value of four parasite species relative to the season. * *p*-value < 0.05; **** *p*-value ≤ 0.0001.

| Biguglia | | | | |
|---|---|---|---|---|
| Season | Spring | Summer | Fall | Winter |
| Number of eels | 20 | 20 | 19 | 19 |
| Copepoda *Ergasilus gibbus* | 12.0–100.0 (12.0) | 41.4–100.0 (41.4) * | 14.2–73.7 (10.5) | 32.4–84.2 (27.3) |
| **Urbino** | | | | |
| Season | Spring | Summer | Fall | Winter |
| Number of eels | 18 | 20 | 20 | 14 |
| Digenea *Deropristis inflata* | 0.9–22.2 (0.2) | 0.0–0.0 (0.0) | 13.7–55.0 (7.5) | 85.4–78.6 (67.1) **** |
| Acanthocephala *Southwellina hispida* | 5.7–11.1 (0.6) | 40.8–55.0 (22.4) | 53.5–55.0 (29.4) * | 0.0–0.0 (0.0) |

In Urbino, *B. anguillae* abundance was at its lowest in winter and reached its highest in summer, with prevalence that remained comparable throughout the year. *Deropristis inflata* reached both its highest abundance and prevalence during winter, but it was not detected during summer. This species had a significant indicator value for winter, as both specificity and fidelity were markedly higher than during any other season. The abundance of *L. musculus* was the highest during fall, and both abundance and prevalence were at their lowest during spring. These three Digenea inhabiting the digestive system thus show seasonal dynamics with asynchronous maxima. *Southwellina hispida* was not detected during winter but was present in spring, though with low abundance and prevalence, which both rose in summer and fall. The indicator value of this species was significant during fall, as its specificity was the highest during this season, while fidelity was intermediate. The abundance and prevalence of *A. crassus* were quite low all year round, and even null in fall. *Contracaecum* sp. reached its highest in summer and its lowest in winter both in abundance and prevalence (Table 2). Indicator values were not calculated for seasons in freshwater, as these sites were only sampled once.

*3.4. Influence of Silvering Stage and Age on Parasite Communities in Brackish and Freshwater Sites*

Concerning the silvering stage, the Cestoda *Proteocephalus macrocephalus* (Creplin, 1825) Nufer, 1905 was found to be an indicator (*p*-value < 0.05, Table 6) of the FII silvering stage (female, growth phase) in Biguglia. The specificity was very high, but it was not at the maximum, as the parasite was also found in the I silvering stage (undifferentiated phase), and fidelity was quite low. *Contracaecum* sp. was found to be an indicator of the FIII silvering stage (pre-migrant) in Urbino, with quite high specificity and maximum fidelity. No parasite was found to be an indicator of any silvering stage in freshwater sites.

**Table 6.** Specificity (%), fidelity (%), and indicator value of two parasite species relative to the silvering stage of the eel. * *p*-value < 0.05.

| Biguglia | | | | | |
|---|---|---|---|---|---|
| Silvering Stage | I | FII | FIII | FV | MII |
| Number of eels | 38 | 6 | | | 34 |
| Cestoda *Proteocephalus macrocephalus* | 2.6–2.6 (0.0) | 97.4–16.7 (16.2) * | - | - | 0.0–0.0 (0.0) |
| **Urbino** | | | | | |
| Silvering stage | I | FII | FIII | FV | MII |
| Number of eels | 38 | 21 | 4 | 2 | 7 |
| Nematoda *Contracaecum* sp. | 6.6–50.0 (3.3) | 6.8–85.7 (5.8) | 66.5–100.0 (66.5) * | 7.6–100.0 (7.6) | 12.5–71.4 (8.9) |

Concerning individual age, *A. crassus* was found to be an indicator of the youngest eels (age 5–6 a) (*p*-value < 0.05, Table 7) in Urbino. The specificity was quite high, but not at the maximum, as the Nematoda was found in all classes, and the fidelity was quite low. *Contracaecum* sp. was an indicator of the oldest eels (>10 a) in the same lagoon. The specificity was intermediate, but it was higher than in any other age class; the fidelity was at the maximum. In rivers, *Pseudodactylogyrus* spp. and *P. tenerrima* were both found to be indicators of the oldest eels. The specificity of both species and the fidelity of

*Pseudodactylogyrus* spp. were quite high; the fidelity of *P. tenerrima* was intermediate. No parasite was found to be an indicator of any age class in Biguglia lagoon.

**Table 7.** Specificity (%), fidelity (%), and indicator value of two parasite species relative to the age of the eel. * *p*-value < 0.05.

| | Urbino | | |
|---|---|---|---|
| **Age** | **4–6 a** | **7–9 a** | **≥10 a** |
| Number of eels | 11 | 47 | 14 |
| Nematoda | | | |
| *Anguillicola crassus* | 75.5–27.3 (20.6) * | 7.6–4.3 (0.3) | 16.9–14.3 (2.4) |
| *Contracaecum* sp. | 29.3–54.5 (16.0) | 15.2–59.6 (9.1) | 55.4–100.0 (55.4) * |
| | Rivers | | |
| **Age** | **4–6 a** | **7–9 a** | **≥10 a** |
| Number of eels | 30 | 97 | 30 |
| Monogenea | | | |
| *Pseudodactylogyrus anguillae + P. bini* | 17.0–76.7 (13.1) | 22.6–67.0 (15.2) | 60.3–76.7 (46.3) * |
| Nematoda | | | |
| *Paraquimperia tenerrima* | 13.7–23.3 (3.2) | 20.4–21.6 (4.4) | 65.9–36.7 (24.2) * |

## 4. Discussion

### 4.1. Low Parasite Diversity in Mediterranean Islands

The sedentary lifestyle of *A. anguilla* allows the heterogeneity of the local parasite community to be maintained [44]. Parasite communities of *A. anguilla* seem to reflect the habitat in which the host dwells. Despite similarities in specific composition, the parasite communities from the three types of habitat studied were indeed well-differentiated and support the idea of a salinity dependence of parasite species' composition in *A. anguilla*. Comparing the diversity index with other results obtained from the *A. anguilla* distribution range is made challenging, because in most studies, these indices were calculated at the component community level, whereas they were calculated at the individual level in the present study in order to be able to retain information about inter-individual variability. However, the Shannon indices calculated in the present study seem to be among the lowest, and the Berger–Parker dominance indices are among the highest observed in Europe compared to other studies [9,15,45–49]. The brackish Digenea suite was species-poorer than what might be expected, with the notable absence of certain species, such as *Helicometra fasciata* (Rudolphi, 1819) Odhner, 1902, commonly found in South European ecosystems [50–53]. This is surprising, as this parasite is known to be present along the Corsican coasts in marine fish, such as *Labrus merula* Linnaeus, 1758 and *Symphodus rostratus* (Bloch, 1791) [54,55]. Digenea diversity was also low in Sardinia [12]. In freshwater habitats, parasite communities are frequently species-poor, exhibiting low diversity and high dominance [50], which is in agreement with the present findings. However, freshwater habitats are frequently reported to be dominated by Acanthocephala [48,56–59], a taxon that was completely absent in Corsican rivers.

The low parasite diversity observed in the present study and the notable absence of parasites common in continental Europe could be related to the relative isolation of the island of Corsica in the Mediterranean, and it could also be a general trend in isolated Mediterranean islands. Islands are known to be species-poorer than mainland areas [60,61]. Further studies should be conducted in other Mediterranean islands in order to confirm this hypothesis.

*4.2. Habitat Specificity of A. anguilla Parasites*

4.2.1. Parasites in Eels from Brackish Coastal Lagoons

The Copepoda *E. gibbus* was only found in Biguglia lagoon and in the near-to-sea Abatesco freshwater site (8 m altitude, 2 km distance to sea). This is consistent with results from across the host's distribution range, with reports of this parasite in freshwater sites and in brackish sites [9,11,62–65]. In Southern France and Tunisia, *E. gibbus* was reported from waters that never reach a high salinity [66].

The three Digenea *B. anguillae*, *D. inflata*, and *L. musculus* are a typically marine suite already recovered from Corsica [10,11] as well as from other Mediterranean lagoons, from Italy [12,50,51,67], and from other places in the distribution range of *A. anguilla* (e.g., Spain and Iceland) [9,45,53,68,69]. The life cycle of *B. anguillae* was studied by Gargouri-Ben Abdallah and Maamouri [70], who showed that the parasite could complete its cycle using the Bivalvia *Abra tenuis* (Montagu, 1803) as the first intermediate host and the Mediterranean banded killifish *Aphanius fasciatus* (Valenciennes, 1821) as the second intermediate host. *Aphanius fasciatus* can be found in both Biguglia and Urbino lagoons [32,71], but not *A. tenuis*. Other species of *Abra* have been reported from both Biguglia (*Abra segmentum* (Récluz, 1843)) and Urbino (*A. segmentum* and *Abra alba* (W. Wood, 1802)) [32]. It thus seems that the presence of *B. anguillae* in Corsican lagoons cannot be explained by the presence/absence of its intermediate hosts, at least without density data, which are not available. Known intermediate hosts for *D. inflata* are, first, the marine Gasteropoda *Bittiolum alternatum* (Say, 1822), and then annelids, such as *Alitta virens* (M. Sars, 1835) and *Hediste diversicolor* (O.F. Müller, 1776) [20,72]. *Hediste diversicolor* was reported from both Biguglia and Urbino, but not *B. alternatum*. However, another Gasteropoda of the family Bittinae, *Bittium reticulatum* (da Costa, 1778), was reported from Urbino lagoon [32]. This is surprising, as *D. inflata* was the only Digenea found in both Corsican lagoons in the present study and in 2009–2013 [11]. This could be explained by the fact that no recent data are available concerning the Mollusca and Annelida fauna of Corsican lagoons, thus preventing us from knowing whether any long-term variation of these communities has occurred, or whether this is due to an unreported ability of the parasite to accept a wider range of intermediate hosts. The *Lecithochirium musculus* life cycle remains unelucidated, with the only information available being that *Lecithochirium* species are typically acquired through the consumption of small rock-pool fishes, such as gobies and labrids [73], which are present in both lagoons [32]. As Dezfuli et al. [50] have already pointed out, there are few long-term datasets on *A. anguilla* parasite communities with which to compare the present findings. They reported that helminth parasites' richness and diversity did not change significantly over 16 years in Comacchio lagoon (Italy), and they also reported a conservation of the same dominant species over the period despite changes in the community composition. In the present study, the dominant species also remained the same for both lagoons. In Comacchio lagoon, much of the stability could be linked to the dominance of a suite of marine Digenea [50], which is also the case in Urbino lagoon. The authors concluded that this could be a general characteristic of coastal lagoons, as a similar suite dominates the communities in other Adriatic and Tyrrhenian lagoons [51,67].

The Acanthocephala *S. hispida* was found in both lagoons, but with markedly higher prevalence and abundance in Urbino. This parasite was already reported in Italy: once as an adult in one of its definitive hosts, the great cormorant *Phalacrocorax carbo* (Linnaeus, 1758), and as cystacanths in fish hosts, including *A. anguilla* [12,74]. This is thus the second time the presence of this parasite's larval stage is noted in *A. anguilla*. The relative proximity between these two reports, both geographically and in terms of habitat, may be noted, as Culurgioni et al. [12] found it in the polyhaline-to-euhaline Santa Gilla lagoon in Southern Sardinia. Definitive hosts for this parasite are fish-eating birds, such as *P. carbo*, which is known to frequent both Biguglia and Urbino. *Anguilla anguilla* could be a paratenic host for *S. hispida*, as this parasite is known to have a Decapoda intermediate host and to accept several vertebrate species (e.g., freshwater and brackish fish and frogs) as paratenic hosts [75].

*Contracaecum* sp. larvae dominated in Urbino lagoon, where they had higher prevalence and abundance compared to Biguglia lagoon and freshwater sites in Corsica. *A. anguilla* is an intermediate or paratenic host for *Contracaecum* species, and it acquires the larval stage through consuming an intermediate invertebrate host. Adult *Contracaecum* are found in the digestive system of fish-eating birds or marine mammals [62,76]. *Contracaecum* sp. is found in a wide range of salinity throughout the *A. anguilla* distribution range, but it is more often reported from brackish and euhaline conditions [9,11,50,51,53,67]. IndVal analysis showed *Contracaecum* sp. to be an indicator of FIII silvering stage eels (pre-migrant females) in the present study and in silver eels in 2009–2013 [11], of the oldest sampled eels (at least 10-year-old eels) in the present study, and of 50–60 cm eels [11]. This observation of more *Contracaecum* sp. encysted larvae in older and larger eels is likely due to an accumulation of this parasite through a longer exposure time and an enhanced consumption of intermediate hosts [77]. It should, however, be noted that classic visual examination was carried out to detect the parasites, while no complementary methods, such as the digestion or incubation methods [78,79], were used. Moreover, Shamsi et al. [80] showed that *Contracaecum* larvae could be deeply embedded in intestinal tissues and thus evade detection until emergence upon exposition to a heat source. The reported prevalence and/or abundances could thus be underestimated.

### 4.2.2. Parasites in Eels from Freshwater Sites

The Nematoda *P. tenerrima* is known to be a freshwater parasite [76]. It was found only in freshwater sites in the present study and throughout the distribution range of its host [16,51,62,81,82], with few exceptions [9,10]. Shears and Kennedy [83] elucidated the life cycle of *P. tenerrima* by showing that the minnow *Phoxinus phoxinus* (Linnaeus, 1758) is an obligatory intermediate host for this parasite, and that *A. anguilla* is infected through the consumption of this intermediate host. *Phoxinus phoxinus* is not native to Corsica, but it has been introduced into the freshwater habitats of the island, thus presumably allowing this parasite's life cycle to be completed. *Paraquimperia tenerrima* could even have been co-introduced with *P. phoxinus*, but no data on the parasitofauna of freshwater eels anterior to *P. phoxinus* introduction in Corsica are available to test this hypothesis.

### 4.3. Invasive Species

Three invasive species were the most widespread parasites of *A. anguilla* in Corsica, with *Pseudodactylogyrus* recovered from all sites except Urbino lagoon and *A. crassus* present in the entirety of the sampled sites. Both *Pseudodactylogyrus* and *A. crassus* are considered invasive parasites [6,31,84]. These parasites meet the criteria of invasive species as they: (1) are non-native organisms introduced outside of their native range; (2) establish self-sustaining populations that spread beyond their area of introduction; and (3) have deleterious impacts on either the environment, the economy, or human health [85]. Both helminths were accidentally introduced from their native range in Asia to Europe despite early warnings concerning the high susceptibility of *A. anguilla* [86].

### 4.3.1. *Pseudodactylogyrus anguillae* and *P. bini*

*Pseudodactylogyrus anguillae* and *P. bini* were first reported in Europe in an eel farm in the Soviet Union [25]. In France, the parasites have been reported since 1985 in rivers [87]. *Pseudodactylogyrus* has now spread throughout the distribution range of *A. anguilla*, except for Iceland (Figure 3a), and it has also been reported in North America on the American eel *Anguilla rostrata* (Lesueur, 1817) [88] and in South Africa in the giant mottled eel *Anguilla marmorata* (Quoy and Gaimard, 1824) [89]. *Pseudodactylogyrus* is known to hamper the commercial production of *A. anguilla* [86]. The parasite causes extensive hyperaemia on the gills, increased mucus secretion and damage to the gill structures, decreased food intake, and lethargy; untreated infection can lead to up to 90% mortality in aquaculture [31,84]. Although they may cause tissue damage, impaired respiration, and signs of stress in wild eel, they do not seem to be responsible for mortality or have any effect on migration [27,29].

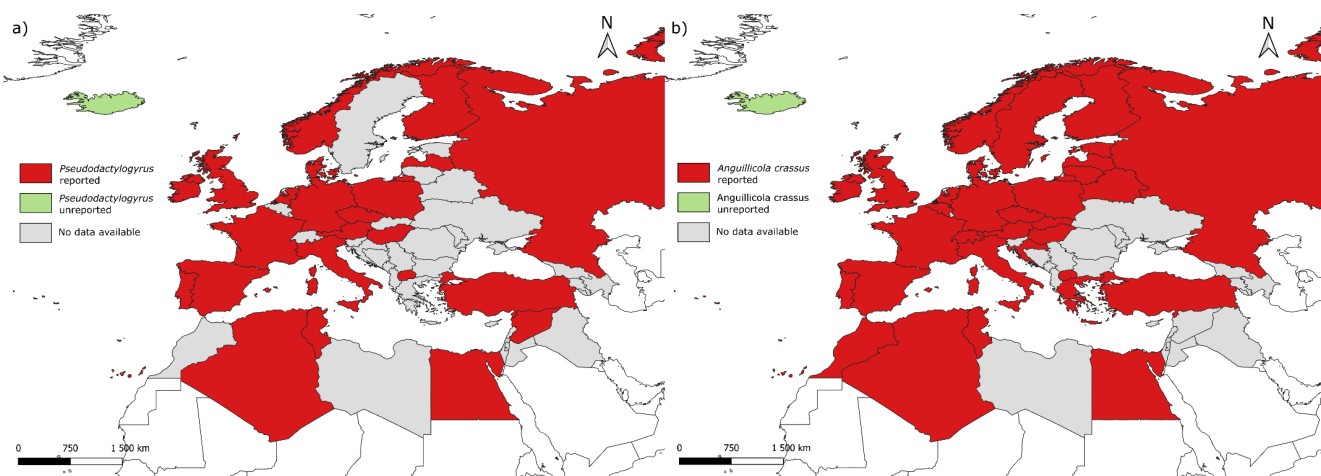

**Figure 3.** Reports of (**a**) *Pseudodactylogyrus* spp. and (**b**) *Anguillicola crassus* according to the existing literature on *Anguilla anguilla* parasites.

Both *Pseudodactylogyrus* were found in freshwater sites and Biguglia lagoon, but they were absent in the more saline Urbino lagoon. This distribution is consistent with parasite community data from the distribution area of *A. anguilla*, where these Monogenea are mainly found in freshwater and brackish fjords, lagoons, and estuaries (e.g., [14,62,90–92], with the exception of some Mediterranean lagoons where the salinity can reach a relatively high value [51,68]. In Germany, this parasite could not be found in the North Sea marine environment [9]. The general trend of *Pseudodactylogyrus* infections is an increase of the infection with a larger host area [93–95]. The same trend was surprisingly only found in freshwater sites in Corsica in the present study, with *Pseudodactylogyrus* species being an indicator of older-than-10-year-old eels. *Pseudodactylogyrus* species showed the highest abundance of infection during summer and fall, which is congruent with the observation from the period of 2009–2013 [14] and observations from other European sites [94,96]. *Pseudodactylogyrus* species being ectoparasites, it is likely that the observed seasonality results from the environmental preferences of these parasites. Biguglia is a shallow lagoon with a maximum depth of 1.8 m, and the water temperature is strongly dependent on the air temperature [32], reaching its highest in summer.

*Pseudodactylogyrus* abundances in Biguglia lagoon were higher in 2021–2022 compared to 2009–2012 for all seasons (Wilcoxon–Mann–Whitney tests, *p*-value < 0.05, Figure 4, Tables 2 and S1). The prevalence of infection also increased between the two periods, as we found all investigated individuals to be infected by the parasite, which was not the case in 2009–2012. *Pseudodactylogyrus* was reported as soon as 1999 in Biguglia [92], with markedly lower abundance and prevalence, and it was already noted as an indicator species in this lagoon during the period of 2009–2013. No data could be found on the long-term evolution of *Pseudodactylogyrus* infection in its invasive distribution range, but the present study clearly showed a sharp increase in both abundance and prevalence in Corsica over 10 years, which is consistent with its invasive character.

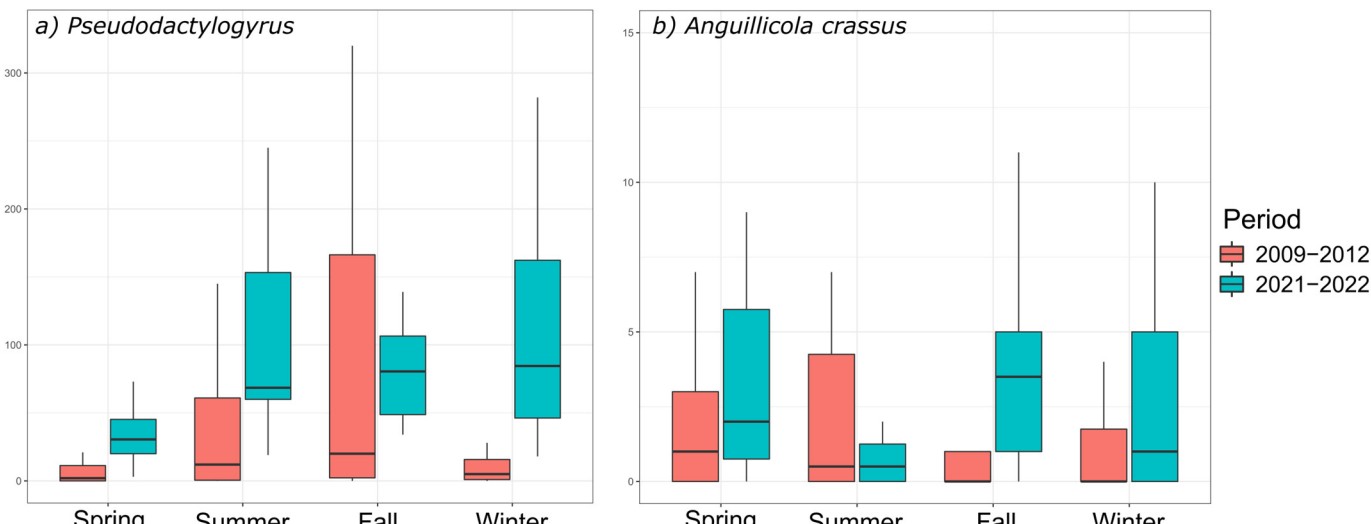

**Figure 4.** Abundance of infection for (**a**) *Pseudodactylogyrus anguillae* and *P. bini* and (**b**) *Anguillicola crassus* for the brackish Biguglia lagoon for two periods. For the period of 2009–2012, data from several sampling years were pooled by season.

### 4.3.2. *Anguillicola crassus*

*Anguillicola crassus* was introduced in Germany around 1892 through the importation of live eels from Taiwan [6] and then dispersion across its new host's distribution range, where it is now found except for in Iceland (Figure 3b). The parasite has also been reported from North America on *A. rostrata* [97]. *Anguillicola crassus* is frequently cited as one of the possible factors involved in the decline of *A. anguilla*, because it infects the swim bladder and is thus feared to hamper the transatlantic spawning migration of its host by damaging the functionality of this essential hydrostatic organ [29]. This parasite was shown to impair the silvering-related enhancements of the reactive oxygen species (ROS) defense capacity in the swim bladder tissues [98] and to reduce the mechanical integrity of the swim bladder [99]; it also seems to cause a decrease in macrophage phagocytic response that could result in higher susceptibility to other pathogens [100]. Infected eels show a higher stress response to hypoxia than non-infected eels [101]. However, according to Taraschewski [6], all eel species undergoing intensive human management (*A. anguilla*, *A. japonica*, and *A. rostrata*) saw their abundance decline since the mid-1970s independently from the arrival and persistence of *A. crassus*.

*Anguillicola crassus* was found in all investigated sites in Corsica, both in freshwater sites and brackish lagoons. Its high tolerance towards ecological factors, such as temperature and salinity, is one of the reasons proposed to explain its success as an invasive species [6]. Another reason is its acceptance of a wide range of Copepoda and Ostracoda intermediate hosts [102,103]. Despite this high tolerance towards habitat conditions, *A. crassus* is more commonly found in low-to-moderate salinity habitats than in euhaline conditions, frequently with a higher abundance. In Algeria, both the prevalence and abundance of *A. crassus* were markedly higher in the freshwater Oubeïra Lake than in the polyhaline-to-euhaline Mellah lagoon [104], where those parasite indices had values similar to those found in Urbino lagoon in the present study. Abundances were also quite low in Spanish and Italian brackish Mediterranean lagoons, in Kattegat, and in the marine locality of Helgoland [9,51,67,68,105], whereas they tended to be higher in low-salinity habitats, such as German lakes and rivers, Lake Ohrid in Macedonia, and Belgian rivers [9,49,106–108]. Several studies have pointed out either one or two annual maxima in abundances and/or the prevalence of *A. crassus*: some showed the highest abundance and/or prevalence in fall [106,109,110], while others showed the highest abundance and/or prevalence in summer [77,111,112]. The present study showed a maximum prevalence in spring and fall, and a slightly higher abundance in spring. Taraschewski [111] has already

proposed that high summer and low winter temperatures could prevent the progress of this parasite's life cycle, or that this variability could be linked to a critical period when the eels lower their food intake, and, thus, the uptake of infective stages through the consumption of intermediate hosts.

*Anguillicola crassus* abundances were higher in 2021–2022 than in 2009–2012 for all seasons (Wilcoxon–Mann–Whitney tests, *p*-value < 0.05), except during summer (Figure 4, Tables 2 and S1). The prevalence of infection followed the same trend. The presence of this Nematoda in Biguglia lagoon was first reported in 1999 [92], with similar prevalence and abundance, and this species was already an indicator of Biguglia in 2009–2012 [11]. According to Wielgoss et al. [113], results gathered from European sites showed stabilization over time, and even a slight decline in the abundance and intensity of infection by Nematoda parasites, which could reflect an increase in resistance towards these parasites in the long term. In Vaccarès lagoon, Fazio et al. [18] showed a stabilized infection by *A. crassus* between 1997 and 2004. The present results are congruent with this observation only in Urbino, where infection parameters were low in 2009–2012 and remain low today. In contrast, both the prevalence and abundance of *A. crassus* significantly rose between 2009–2012 and 2021–2022 in Biguglia. An increase in infection by this parasite was also observed between 1988 and 2004 in Mauguio lagoon (Southern France) [18,114]. The apparent absence of *A. crassus* in fall 2021 in Urbino is certainly a sampling artefact related to the low prevalence of infection in this lagoon.

### 4.3.3. Implications for the Management of the European Eel

Neither the present study nor Jakob et al. [9] could show an influence of the parasitic load of *A. crassus* or *Pseudodactylogyrus* on the body condition of eels, and no sign of degraded health was observed in the samples. However, given the strong load found on some individuals (up to 1200 individual *Pseudodactylogyrus* and 41 *A. crassus* on an eel) and the well-known pathogenicity of these parasites, an adverse effect on the fitness of strongly infected hosts cannot be excluded. As the conservation of *A. anguilla* populations is dependent on the spawners' health and fitness for their transatlantic migration, there is a need for efficient management plans.

The reasons given to explain the absence of both invasive parasites in Iceland are the low winter temperatures that would hamper the parasites' development, and the geographical isolation and absence of permission to import live eels [45], highlighting the importance of strict and early management measures to avoid biological invasions. As the distribution area of *A. anguilla* is already being invaded by *Pseudodactylogyrus* and *A. crassus*, management measures should take into account the environmental preferences of these invaders. In Corsica as well as in the rest of the *A. anguilla* distribution area, the prevalence and parasite load of both invaders are markedly lower, or even non-existent, in high-salinity habitats. Jakob et al. [9] have already pointed out that eels residing in high-salinity habitats were not at risk regarding infection by these parasites and thus could be advantaged in reaching their spawning ground with better health condition and fitness. Consequently, restocking freshwater sites with glass eels caught in estuaries (a common restocking practice in France and in Europe) may have reduced the number of eels that would have stayed in the marine environment, where they would have been protected from infection by high salinity. Therefore, future eel management measures should avoid restocking freshwater habitats with wild-caught glass eels.

### 5. Conclusions

In conclusion, the extensive survey of macroparasites in *A. anguilla* in the little-studied island of Corsica revealed the presence of 19 species. The first-time study of inland water of this isolated territory allowed us to fill in some of the gaps concerning diseases and parasites of a strongly declining, prized, and exploited species, and to reveal the presence of three invasive parasite species (*P. bini*, *P. anguillae*, and *A. crassus*) in freshwater habitats. The comparison of parasite communities between brackish and freshwater habitats showed

the clear preference of the parasites towards the host's habitat and salinity. Seasonal variations were shown. Comparing the results obtained in the present study with data from across the European eel's distribution range highlighted the relatively low parasite diversity in Corsica and the lack of a common parasite as in mainland Europe, such as Acanthocephala. Determining the silvering stage and the individual age of eels allowed us to bring to light the preference of several species towards biotic parameters. The study of long-term variations in the coastal brackish lagoons of Corsica revealed the increase of both the prevalence and abundance of the three invasive parasite species in the last decade. Future management measures should take into account the environmental preferences of pathogenic parasites.

**Supplementary Materials:** The following supporting information can be downloaded at: https://www.mdpi.com/article/10.3390/fishes8070375/s1, Table S1: Composition of parasite communities of *Anguilla anguilla* in Corsican lagoons in 2009–2013, with information on prevalence, mean abundance (±standard deviation), mean intensity (±standard deviation), and diversity indices: Shannon's diversity index and Berger-Parker's dominance index. Data obtained from Filippi et al. (2013) [13]; in this paper, cyst of Acanthocephala were not identified, but as they were found in the same habitat (Urbino Lagoon) as those found in the present study, they were supposed to be the same species (*Southwellina hispida*).

**Author Contributions:** A.E., J.F. and Y.Q. conceived and designed this study. A.E., Q.G., C.G., P.-J.A. and C.A. conducted the fieldwork in freshwater habitats. A.E., Q.G., C.G. and J.-J.F. participated in fieldwork in brackish lagoons. A.E., Q.G., C.G. and Y.Q. conducted parasitological examination. A.E., C.G., R.M. and E.D. conducted otolithometry analysis. A.E. and C.G. performed statistical analyses. All authors have read and agreed to the published version of the manuscript.

**Funding:** The present study was partially funded as a doctoral fellowship of the University of Corsica Pasquale Paoli and the *Culletivittà di Corsica* granted to A.E. This research is part of the GERHYCO interdisciplinary project dedicated to water management, ecology, and hydro-ecosystem services in an insular context, and it was financially supported by the *Culletivittà di Corsica*.

**Institutional Review Board Statement:** The capture of eels was conducted in accordance with French legislation, under *Arrêtés préfectoraux* delivered by the *Préfectures* of Haute-Corse and Corse-du-Sud (ref. N°2B-2021-06-11-00001 of 11 June 2021 and N°2A-2021-06-29-00002 of 29 June 2021) by regulatory trained agents. Lagoon samples were bought from professional fishermen. Euthanasia was performed in accordance with French legislation (*Arrêté du 1er février 2013 fixant les conditions d'agrément, d'aménagement et de fonctionnement des établissements utilisateurs, éleveurs ou fournisseurs d'animaux utilisés à des fins scientifiques et leurs contrôles*, available at https://www.legifrance.gouv.fr/loda/article_lc/LEGIARTI000027038776). No experiment was conducted on live animals.

**Data Availability Statement:** The authors confirm that the data supporting the findings of this study are available within the article and in its supplementary material. Raw data that support the findings of this study are available from the corresponding author, upon reasonable request.

**Acknowledgments:** The authors are grateful to Pierre Planet and Louis Tarallo and their teams, the fishermen of Biguglia lagoon and Urbino lagoon, respectively, for providing the eels. We also thank our colleagues of the Fédération de la Corse pour la Pêche et la Protection du Milieu Aquatique, Alain Martin, Joseph Canale, and Olivier Saget, and the Office Français de la Biodiversité field agents.

**Conflicts of Interest:** The authors declare no conflict of interest.

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
