# Peer review of "Macroparasite Communities with Special Attention to Invasive Helminths in European Eels Anguilla anguilla from Freshwaters and Brackish Lagoons of a Mediterranean Island"

_fishes, doi:10.3390/fishes8070375_

Round 1

Reviewer 1 Report

The European eel is a strongly declining species, as well as a fish that is prized and exploited by the fisheries of Europe and North Africa, both in fishing and in farming. This presents a great challenge to the authorities in charge of defending the eel because they are faced with difficult environmental tests against unstoppable natural processes, and socio-economic arguments.

One of the factors considered to be responsible for the decline of the eel and on which research attention should be focused are diseases and parasites, in particular the evolution of their incidence on eel subpopulations and on farms in recent years. The present work contributes to this aspect, highlighting for the first time some of the most dangerous parasite species, and comparing their parasitological indices over time and in different environmental contexts in a region that has been little studied previously, such as Corsica's inland waters.

The MS is well presented and results are clear, and the management implications of the results are correctly highlighted in the discussion.

As a general suggestion for the MS, please write the scientific names in full and the authorship (check https://www.marinespecies.org/) when a species is mentioned for the first time (in both abstract and text), and each time a scientific name opens a sentence.

Last, a photograph of at least one of the most representative parasite species treated, such as Anguillicola crassus, would have been appreciated. You may consider it as a mere suggestion.

Reviewer 2 Report

Thank you for inviting me to review this manuscript. Authors conducted an extensive survey of macroparasites in European eels in Corsica, France, and identified nineteen parasites. The study revealed preferences of parasites for the eel's habitat and salinity, as well as seasonal variations. This study aligns with the scope of Fishes.

I have the following questions and comments:

I could not see animal ethic approval details for the study.

Line 46: The sentence should be clarified to specify that non-parasitic invasive species are being referred to.

Line 50: Change "a lot of" to "the."

Line 51: Change "recent" to "more advanced and updated data."

Lines 52 to 81: These lines read like the Results section and should be moved accordingly. Alternatively, provide a table summarizing the previous reports and comment on the table. Otherwise, it may appear as if a systematic review was conducted.

Lines 81 to 95: This section sounds like the Discussion and should be moved accordingly.

Table 2: For "Contracaecum sp.," please note that "sp." should not be italicized.

Tables 2 and 3: It is important to provide the total number for each parameter in order to provide a better understanding of prevalence and other quantities.

The authors did not use the ingestion method (Canadian Journal of Fisheries and Aquatic Sciences 1986, 43(7), 1312-1317) or the incubation method (International Journal of Food Microbiology, 2016, 227, 13-16). Therefore, all the reported prevalence findings could potentially be higher than reported. This should be noted in the Discussion.

The first paragraph of the Materials and Methods section of the article published in the Journal of Helminthology (2017, 92(2), 216-222) could be of interest, as the authors discuss the emergence of Contracaecum larvae deeply embedded in the intestinal tissue of the fish.

References: Scientific names should be italicized and consistently follow a specific style.

Throughout the manuscript, the English language could be improved.

Round 2

Reviewer 2 Report

Authors have addressed all my comments

I think it will benefit the authors if they get the manuscript to be edited by a professional editor.